


# Level 1b error budget for MIPAS on ENVISAT

Anne Kleinert[1], Manfred Birk[2], Gaétan Perron[3], and Georg Wagner[2]

[1]Institut für Meteorologie und Klimaforschung (IMK-ASF), Karlsruher Institut für Technologie, Karlsruhe, Germany
[2]Deutsches Zentrum für Luft- und Raumfahrt, Wessling, Germany
[3]ABB Inc, Québec, Canada

**Correspondence:** Anne Kleinert (anne.kleinert@kit.edu)

**Abstract.** The Michelson Interferometer for Passive Atmospheric Sounding (MIPAS) is a Fourier Transform Spectrometer measuring the radiance emitted from the atmosphere in limb geometry in the thermal infrared spectral region. It was operated on-board the ENVISAT satellite from 2002 to 2012. Calibrated and geolocated spectra, the so-called level 1b data, are the basis for the retrieval of atmospheric parameters. In this paper we present the error budget for the level 1b data of the most recent

data version 8 in terms of radiometric, spectral and line of sight accuracy. The major changes of version 8 compared to older versions are also described. The impact of the different error sources on the spectra is characterized in terms of spectral, vertical and temporal correlation, because these correlations have an impact on the quality of the retrieved quantities. The radiometric error is in the order of 1 to 2.4 %, the spectral accuracy is better than 0.3 ppm, and the line of sight accuracy at the tangent point is around 400 m. All errors are well within the requirements and the achieved accuracy allows to retrieve atmospheric

parameters from the measurements with high quality.

## 1 Introduction

The Michelson Interferometer for Passive Atmospheric Sounding (MIPAS; Fischer et al., 2008) is an infrared Fourier Transform Spectrometer (FTS) operating in the spectral range from 685 to 2410 cm$^{-1}$ (about 4.15 to 14.6 µm). It was operated on a sun synchronous orbit on-board the ENVISAT satellite from 2002 to 2012. MIPAS is a limb emission sounder measuring the

atmospheric emission at tangent altitudes from about 6 to 70 km in nominal measurement mode. These measurements allow to retrieve vertical and horizontal (along-track) distributions of temperature and more than 20 trace gases (e.g., Raspollini et al., 2015, 2013; Wiegele et al., 2012; Funke et al., 2009; von Clarmann et al., 2009), including some isotopologues (e.g., Jonkheid et al., 2016; Steinwagner et al., 2007), aerosols (e.g., Günther et al., 2018; Griessbach et al., 2016) and clouds (e.g., Spang et al., 2018; García-Comas et al., 2016). Some species like bromine nitrate, ammonia or sulfur dioxide have been derived for

the first time in the upper troposphere and stratosphere (Höpfner et al., 2009, 2016, 2015). With a mission time of about one decade, it is also possible to derive stratospheric trends of several species (e.g., Valeri et al., 2017; Eckert et al., 2014; Kellmann et al., 2012; Stiller et al., 2012). For some of the species the measurements need a very high precision, accuracy and stability to be useful within the context of current atmospheric research, for instance ozone, methane, water vapor, SF6, and CF4, just to name a few.



The basis for the retrieval are spectrally and radiometrically calibrated and geolocated spectra, the so-called level 1b data. The quality of these data is essential for the quality of the retrieved species, and a good error estimate is required in order to estimate the precision and accuracy of the retrieved atmospheric parameters (see, e.g., Blumstein et al., 2007; Jarnot et al., 2006).

In this paper, we give an overview of the quality of the MIPAS level 1b data. We investigate the different error sources and quantify the precision and accuracy of the calibrated spectra. The different types of errors are discussed, and the errors are characterized in terms of spectral and vertical correlation, as well as correlation in time. The latter is very important for trend analyses. In Sect. 2 and 3, an overview of the instrument and the level 0 to 1b processing is given, respectively. The following sections treat the different error sources and discuss measurement noise (Sect. 4), radiometric accuracy (Sect. 5 and 6), spectral

accuracy (Sect. 7), and line of sight accuracy (Sect. 8). All error sources are summarized in Sect. 9 and they are characterized in terms of spectral, vertical, and temporal correlation.

## 2   The MIPAS instrument

The heart of the instrument is a Michelson-type interferometer with two input and two output ports. It allows to measure two-sided interferograms with a Maximum Optical Path Difference (MOPD) up to $\pm\,20\,\mathrm{cm}$. One input port receives radiation

from the atmosphere, while the second input port looks at a cold plate of high emissivity cooled at $70\,\mathrm{K}$. Each output port is equipped with four detectors (A1 to D1 and A2 to D2 for the two ports, respectively) covering the spectral range from 685 to $2410\,\mathrm{cm}^{-1}$. The spectral coverage of the individual detectors is different for the two ports (see Fig. 1) in order to ensure full spectral coverage even if one detector fails. The spectra from the 8 detectors are summarized in 5 spectral band (denoted A, AB, B, C, and D in Fig. 1) in the level 1b product. The long wavelength channels A1, A2, B1, and B2 use photoconductive

mercury cadmium telluride (MCT) detectors, while photovoltaic MCT detectors are used in the short wavelength channels C1, C2, D1, and D2.

In nominal measurement mode, the instrument is looking at rearward direction in limb geometry (Fig. 2). The altitude of the tangent point corresponds to the center of the instrument Instantaneous Field Of View (IFOV). The MIPAS IFOV size is 0.0523 deg (in elevation) x 0.523 deg (in azimuth), which is roughly equivalent to 3 km (vertically) x 30 km (horizontally) at

the tangent point.

One interferogram, taken at one tangent altitude, is called a sweep, and a set of sweeps taken at different tangent altitudes is called an altitude scan or simply scan. The movement of the interferometer mirrors changes direction from one sweep to the next. One scan is always composed of an impair number of sweeps (17 in Fig. 2), such that the same tangent altitude is sampled with opposite sweep direction from one altitude scan to the next. The two sweep directions are named forward and

reverse, respectively.

MIPAS is equipped with an internal blackbody. For radiometric calibration, the instrument points towards the internal black-body or into deep space, i.e. at a tangent altitude of about $210\,\mathrm{km}$. The radiometric gain is determined from pairs of blackbody





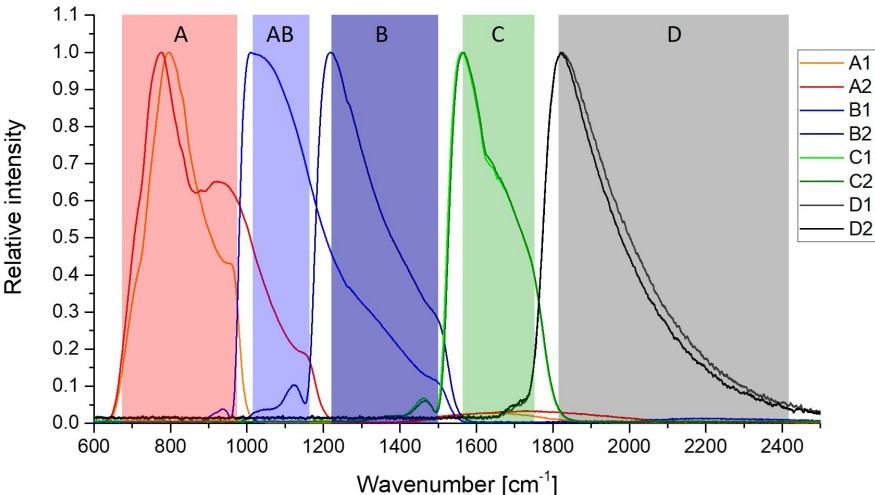

**Figure 1.** Spectral channels and resulting spectral bands of MIPAS. The curves show unfiltered spectra of blackbody measurements in relative units. The maximum of each channel is scaled to 1. Band A is composed of channels A1 and A2, AB of B1, B of B2, and C and D of C1 and C2 and D1 and D2, respectively. The colored boxes indicate the spectral coverage of the spectral bands.

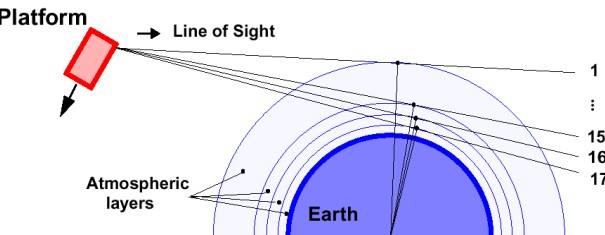

**Figure 2.** Observation geometry of MIPAS (Kleinert et al., 2007).

and deep space measurements, and additional deep space measurements are performed for offset determination. For more details on the instrument, see Fischer et al. (2008).

The MOPD and therewith the spectral resolution has been modified during the mission. From 2002 until March 2004, the full OPD of $\pm 20$ cm, corresponding to a spectral sampling of $0.025$ cm$^{-1}$ was applied for atmospheric measurements.

5  Radiometric calibration measurements were performed with a reduced MOPD of $\pm 2$ cm. Due to increasing anomalies in the velocity of the interferometer drive unit, measurements were suspended in March 2004. In order to minimize the risk of an instrument failure, the following measurements were taken with an MOPD of $\pm 8.2$ cm, identical for atmospheric and calibration measurements. The interferograms are cut to a length of $\pm 8.0$ cm during level 1b processing, corresponding to a spectral sampling of $0.0625$ cm$^{-1}$. After a short test phase in August 2004, measurements were resumed with the reduced

10  MOPD in the beginning of 2005. The shorter measurement time ($1.8$ s instead of $4.5$ s per interferogram) was used to increase



**Table 1.** Radiometric calibration measurements

|  | Full resolution | Optimized resolution |
|---|---|---|
| Spectral sampling of scene measurement | 0.025 cm$^{-1}$ | 0.0625 cm$^{-1}$ |
| Spectral sampling of blackbody and deep space measurement | 0.25 cm$^{-1}$ | 0.0625 cm$^{-1}$ |
| Spectral sampling of blackbody and deep space measurement after on-ground processing | 0.25 cm$^{-1}$ | 0.3125 cm$^{-1}$ (Bands A, AB, B, C) 1.875 cm$^{-1}$ (Band D) |
| Number of coadditions per gain measurement (blackbody and deep space, forward and reverse each) | 300 | 100 |
| Repetition rate of gain measurements | daily | daily |
| Number of coadditions per offset (deep space) measurement (forward and reverse each) | 3 | 6 |
| Repetition rate of offset measurements | ca. 300 s | ca. 700 s |

the vertical sampling from 17 to 27 sweeps per scan in nominal mode, leading to an optimized trade between spectral and spatial resolution. The first measurement period with full spectral resolution is named Full Resolution (FR) mode, while the measurement period from January 2005 to April 2012 is named Optimized Resolution (OR) mode.

The change of the spectral resolution in 2004 also required an adaption of the radiometric calibration measurements. A new
trade between measurement time and noise in the calibration data had to be found (Kleinert and Friedl-Vallon, 2004). Table 1 lists the main characteristics of the calibration measurements for FR and OR measurements.

## 3  Level 1b processing

The measured signal undergoes several processing steps on-board before being sent to ground. The on-board processing includes numerical filtering and decimation, bit truncation and packetizing. Furthermore, the signals of C1 and C2 as well as D1
and D2 are equalized (to match detector responses) and averaged on-board to bands C and D, respectively. The main steps of the level 1b processing are given in short below. A more detailed description of the level 1b processing is given in Kleinert et al. (2007).

**Spike detection and correction:** In case of spikes in the interferograms due to cosmic rays or transmission errors, the affected interferograms are either discarded or the spikes are corrected by a simple correction algorithm. Calibration data with
spikes are discarded, scene data are corrected.

**Fringe count error detection and correction:** In case of fringe count errors during turnaround, the measured interferogram is shifted by an integer number of sampling points. These are corrected by shifting the interferograms back accordingly.

**Detector non-linearity correction:** Due to the non-linear behavior of the photoconductive detectors, their response is dependent on the total photon flux. The non-linearity has been characterized on ground and in flight. A first order correction of
the non-linearity consist in scaling each interferogram according to the incident photon flux. Since the interferogram DC is not measured, the peak-to-peak value of the AC-coupled digitized interferogram ADC$_{max-min}$ is used as a measure for the total



photon flux. The interferograms of the non-linear detectors A1, A2, B1, and B2 are scaled according to their $\text{ADC}_{max-min}$ values before radiometric calibration.

**Radiometric calibration:** A two-point calibration according to Revercomb et al. (1988) is performed using measurements of an internal blackbody and deep space measurements. The radiometric calibration is performed separately for the forward and reverse interferogram sweep direction.

**Spectral calibration:** In order to correct for a drift of the laser wavelength of the reference laser, the spectral axis is scaled by a spectral correction factor. This factor is determined from the spectral position of well-characterized atmospheric lines.

**Geolocation assignment:** The level 1b processor reports with each measured spectrum the geolocation, i.e. the altitude and position over the Earth geoid of the Line Of Sight (LOS) tangent point at the time of the measurement. The geolocation is determined at the measurement time using the satellite attitude and position, the pointing azimuth and elevation mirror angles, scanning mirror non-linearity characterization data, an atmospheric refraction model, and LOS calibration data.

## 3.1 Improvements of the level 1b processing

The general level 1 processing has not changed throughout the mission. In detail, however, the processing has undergone several improvements with new processing versions. In the following, we describe the main improvements for the most recent processing version 8.

**Improved non-linearity characterization:** The analysis of in-flight characterization measurements throughout the mission revealed that the photoconductive detectors are subject to aging. The response is slowly decreasing and with this, the detectors become more linear over time. Moreover, the characterization work has shown that the relation between the size of the interferogram peak ($\text{ADC}_{max-min}$) and the total photon flux is dependent on the instrument temperature and on the degree of ice contamination. In consequence, new parameters for non-linearity correction have been determined from in-flight characterization measurements, depending on time after launch, instrument temperature, and degree of ice contamination. Parameters from in-flight characterization have already been applied to data version 7, but they have again been improved for version 8.

**Improved gain calibration:** Although gain measurements were acquired on a daily basis, the gain function used for radiometric calibration was updated only once per week. The gain variation is usually sufficiently slow that the error introduced by the temporal drift of the gain function is below $1\,\%$. In some situations, however, the gain variation is significantly better captured when using the daily gain measurements (as far as they are available). Therefore it has been decided to use the daily gain measurements for processing version 8.

**Improved spectral calibration:** The spectral calibration factor (SCF) was calculated and updated every four elevation scans. The long-term analysis of the SCF has shown, that the reference laser is much more stable than expected and that the variation of the SCF over time was dominated by the noise of the determination. Therefore the SCF is only updated once per day (together with the radiometric gain function), and mean spectra over one full orbit and the appropriate altitude range are used to determine the spectral calibration factor.

**Improved LOS calibration:** From the LOS calibration data, an annual cycle and negative trend can be deduced. Cycle and trend have been characterized and a corresponding correction has been applied to the tangent altitude information.



## 4   Measurement noise

The measurement noise of the scene spectra is given by the noise equivalent spectral radiance (NESR). It is determined from the imaginary part of the calibrated spectra after high-pass filtering. $NESR_0$ denotes the NESR at zero input radiation to the instrument. The $NESR_0$ has been calculated on ground and in flight. Some examples of NESR spectra together with

5   the requirement are shown in Fig. 3. In order to better compare FR and OR measurements, the $NESR_0$-values of the FR measurements as well as the requirements have been scaled according to the different spectral resolution, i.e. they have been multiplied by $\sqrt{0.025/0.0625}$. In most of the spectral range from about $740\,\mathrm{cm^{-1}}$ to about $2140\,\mathrm{cm^{-1}}$, the NESR is below the requirement and it does not change much over the mission. The NESR for atmospheric measurements at low tangent altitudes is about 20 to 50 % larger than the $NESR_0$, depending on the strength of the atmospheric signal in the different bands (not

10   shown).

The variation of the NESR throughout the mission is shown in Fig. 4. Overall, the variation of the NESR is below 25 %, except for the time period of January to May 2005, where the NESR was increased due to strong ice contamination. The seasonal variation of the NESR as well as the overall small increase over the mission is very well correlated with the instrument temperature (see De Laurentis, 2012, p. 7).

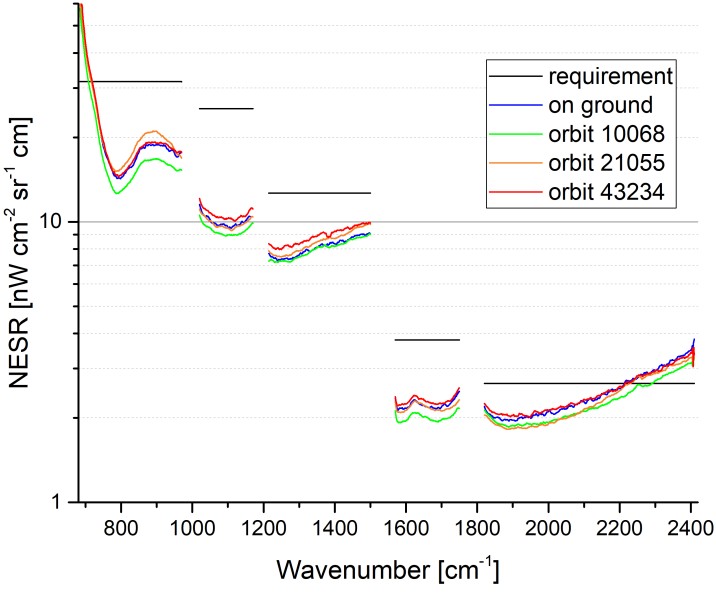

**Figure 3.** $NESR_0$-values on ground and in flight for selected orbits along the mission.




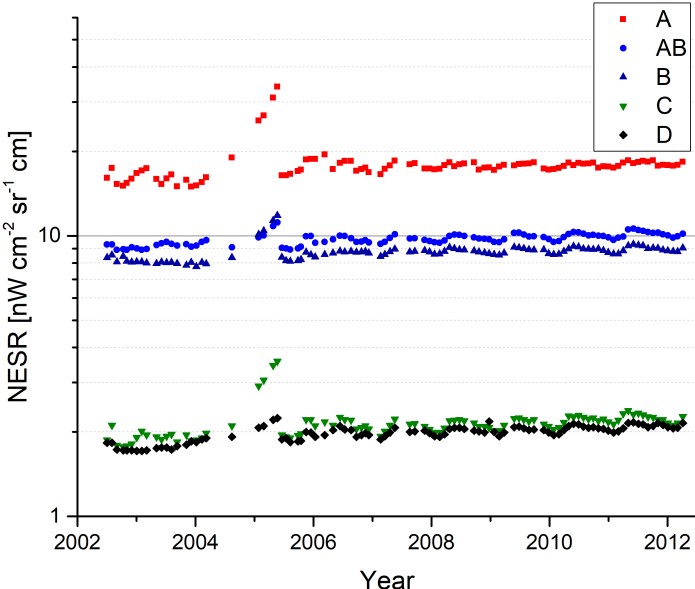

**Figure 4.** NESR-values in the middle of each band throughout the mission. Again, the values measured in FR mode (2002 to 2004) have been scaled to the spectral resolution of the OR mode.

## 5 Radiometric accuracy

The radiometric calibration translates the measured intensities to radiometric units, i.e. to $\mathrm{nWcm}^{-2}\mathrm{sr}^{-1}\mathrm{cm}$. Error sources, which have an impact on the radiometric accuracy are:

- Noise in radiometric calibration measurements
- Temporal variation of the gain function and the instrument offset
- Inaccuracies of the calibration blackbody
- Uncertainty of the non-linearity correction
- Microvibrations
- Pointing jitter

As requirement, the radiometric accuracy shall be better than or equal to the sum of $2\times$ NESR and $5\,\%$ of the source spectral radiance for the non-linear channels, i.e. the spectral range between 685 and $1500\,\mathrm{cm}^{-1}$. For the linear channels, the requirement is the sum of $2\times$ NESR and $2\,\%$ of the source spectral radiance for $1570\,\mathrm{cm}^{-1}$ and the sum of $2\times$ NESR and $3\,\%$ for $2410\,\mathrm{cm}^{-1}$ with a linear increase in this spectral range (Geßner and Fladt, 1995). In fact, a scaling accuracy of $1\,\%$ is desired for band A in order to guarantee an accurate temperature retrieval, but the requirement was relaxed because of the expected uncertainties related to the non-linearity correction.





In this study, the radiometric error is separated into a scaling error and an offset error. The scaling error acts multiplicatively on the spectrum, the offset error acts additively. For all error sources above, the scaling and offset contribution is quantified, and a spectral, temporal and altitude dependency is given, where appropriate. The various error contributions are listed in Tab. 3 in Sect. 9, where a summary of the overall level 1b data accuracy is given.

## 5.1 Noise in the gain measurements

Blackbody and deep space measurements that serve to calculate the gain function show a certain measurement noise. In order to reduce the measurement noise, several consecutive blackbody and deep space measurements are co-added. During commissioning phase, it was verified that these measurements do not contain any highly resolved spectral features. Therefore the spectral resolution of these measurements is reduced in order to further reduce the noise level. The spectral reduction introduces a correlation of the noise between adjacent data points. The gain measurement approach is different for full resolution and optimized resolution measurements. The main characteristics are listed in Tab. 1.

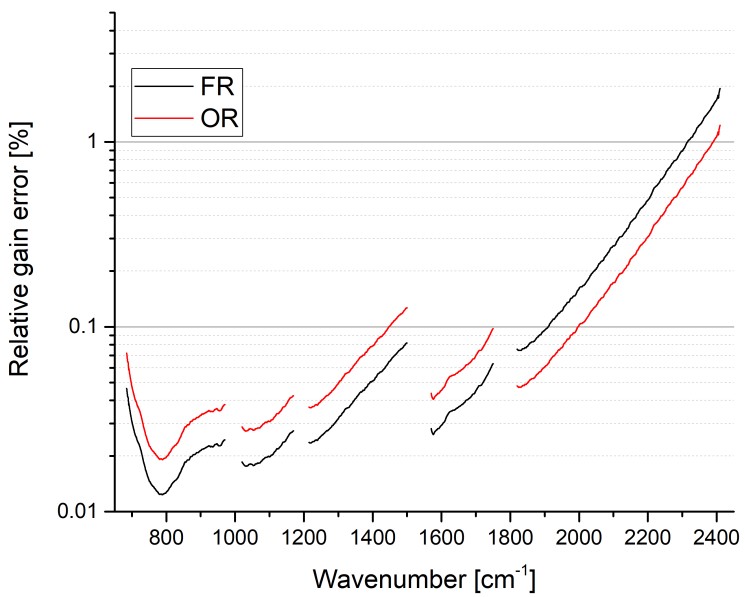

**Figure 5.** Relative gain error due to noise for FR and OR mode in percent. Please note the logarithmic scale.

The gain error due to noise is shown in Fig. 5. Note that this error has a statistical origin, but it acts like a systematic error on the calibrated spectra since the same error due to noise is applied to all spectra of one scan (separate for forward and reverse sweep direction, though) and to a certain number of consecutive scans (usually one day). Furthermore the noise is spectrally


correlated due to the reduced spectral resolution. The 2-$\sigma$-value of the noise amplitude has been used to estimate this systematic error.

## 5.2 Temporal variation of the gain function

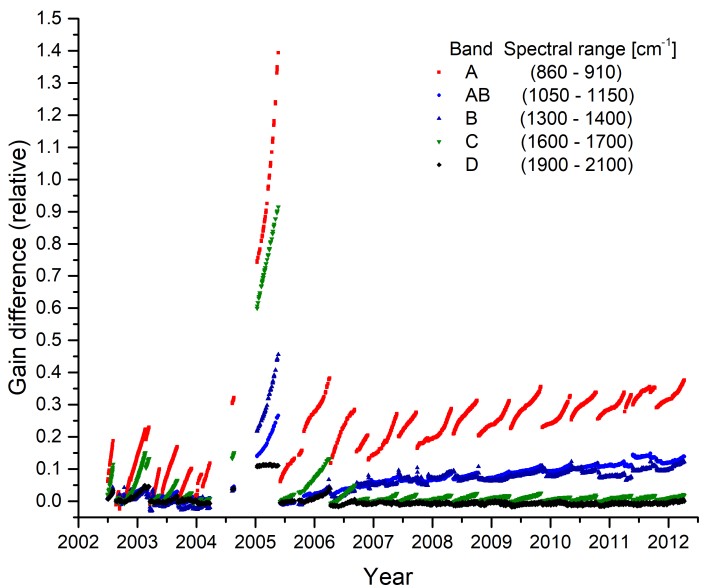

**Figure 6.** Relative gain difference with respect to orbit 2552 of 26 August 2002 in selected spectral regions over the mission.

The gain is determined from a series of blackbody and deep space measurements on a daily basis. In case of measurement interruptions, the time gap may also be more than one day. Figure 6 shows the variation of the gain function in selected spectral regions (one for each band) over the mission. There is a regular increase in the gain function due to ice contamination, followed by a sudden decrease after decontamination. This effect is strongest in bands A and C because of the spectral signature of ice. There is one period with very strong ice contamination between January and May 2005, where no decontamination has been performed.

When looking only at the gain values directly after decontamination, one can observe a continuous increase of the gain function over time in the long wavelength bands A, AB, and B. This is due to detector aging, which affects the photoconductive detectors A1, A2, B1 and B2. Band B and to a lesser extent band AB sometimes show unexplained jumps of up to 2 % in the gain function from one gain measurement to another, often but not always shortly after a decontamination period.




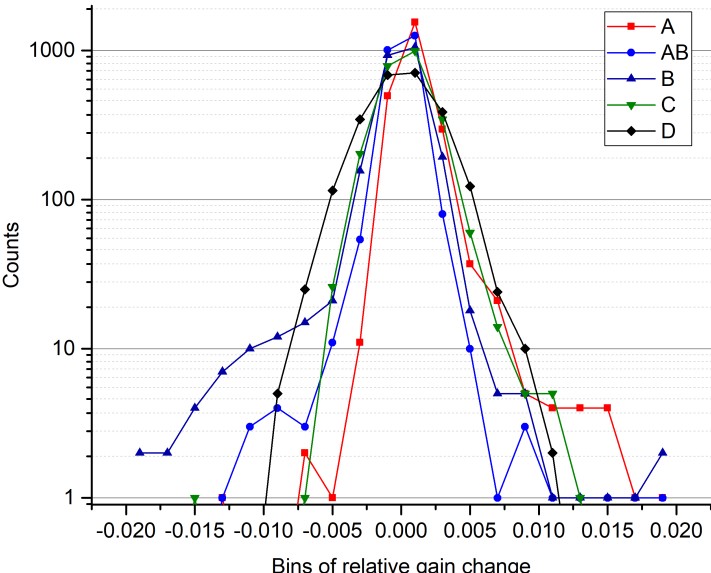

**Figure 7.** Histogram of the relative gain change in the five spectral bands over the mission.

During the level 1b processing, the same gain function is applied to all measurements of that day. If there are days with atmospheric measurements but without gain measurements, the next available gain function in time is applied. Care is taken that the instrument state has not changed between gain and atmospheric measurements, especially in terms of ice contamination.

The variation from one gain measurement to the next is taken as a measure for the uncertainty of the gain calibration. Figure 7

shows a histogram of the gain changes from measurement to measurement in the different bands. Decontamination events have been removed from this statistics. The variation from one gain measurement to the next is below $\pm 1\%$ in more than $98\%$ of the measurements (band A: $98.98\%$, band AB: $99.51\%$, band B: $98.32\%$, band C: $99.67\%$, band D: $99.30\%$). Band A and C show a slight shift to positive values, due to the regular increase of the gain function because of ice contamination. In contrast, the bands AB and B show enhanced values down to -0.5 % (band AB) and to -2 % (band B), respectively, due

to the unexplained gain behavior shortly after decontamination. The FWHM (Full Width at Half Maximum) increases with wavenumber because of the higher measurement noise (relative) at higher wavenumbers.

In order to quantify a typical value for the gain variation from measurement to measurement, the value comprising 95 % of the data is chosen. This leads to a typical gain variation of 0.4 % in bands A, B, and C, 0.3 % in band AB, and 0.6 % in band D. The error is varying slowly with wavenumber, uncorrelated between bands, fully correlated in altitude and fully correlated

in time between two gain measurements (usually one day) but completely uncorrelated from one gain measurement to the next (i.e. on time scales larger than one day or a few days in some situations).



### 5.3 Inaccuracies of the calibration blackbody

The accuracy of the calibration blackbody is limited by the knowledge of the temperature of the cavity, temperature nonuniformities, the quality of the emissivity characterization, and the temperature knowledge of the environment. From on-ground characterization it is estimated to be less than 0.5 % (Châteauneuf et al., 2001). A possible degradation of the blackbody over the mission can be detected by a change in the gain function over all bands. Band D, which is not affected by detector aging, shows a constant gain over the mission. This allows to conclude that the quality of the blackbody is preserved over the instrument lifetime.

### 5.4 Noise in the offset measurements

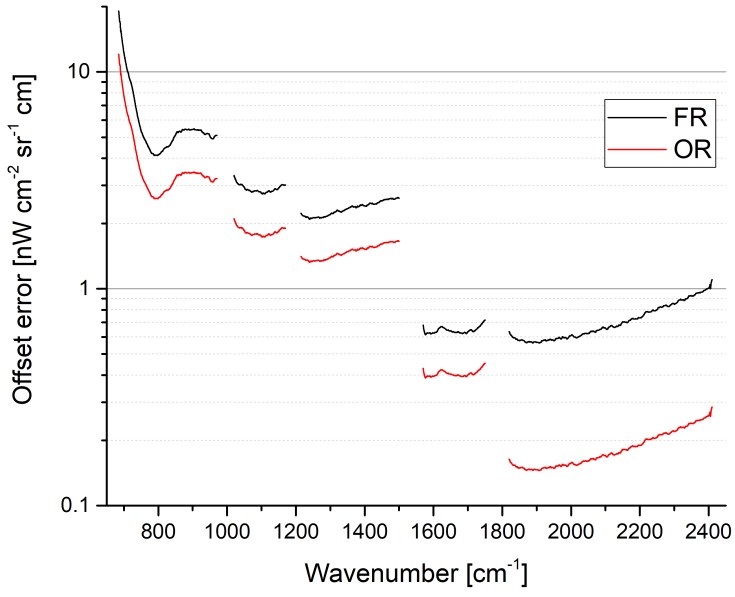

**Figure 8.** Offset error due to noise in the offset measurements.

The offset, which is governed by the instrument self-emission, is determined several times per orbit. The repetition rate as well as the number of coadditions and the spectral resolution are given in Tab. 1. The error due to noise in the offset measurements is shown in Fig. 8. As for the noise in the gain measurements, this error is of statistical origin, but it is systematic in time between subsequent offset measurements, and it is spectrally correlated corresponding to the spectral resolution of the offset measurements. Furthermore the error is constant with altitude (within one limb scan), because the same offset is subtracted from all atmospheric measurements of one scan.



The offset error due to noise in the offset measurements is spectrally correlated within the spectral resolution of the offset, it is constant in time between subsequent offset measurements (i.e. several minutes), and it is vertically constant.

## 5.5 Temporal variation of the instrument offset

The instrument self emission varies slightly along the orbit. This is well captured by the regular offset measurements. Figure 9 shows the offset variation along the orbit for selected wavenumbers in the different spectral bands in November 2003 (FR mode). The variation between two subsequent offset measurements is below $2\,\mathrm{nWcm^{-2}sr^{-1}cm}$ in band A and even lower in the other spectral bands. In OR mode, where the time span between two offset calibration measurements is larger, the variation is below about $4\,\mathrm{nWcm^{-2}sr^{-1}cm}$ and still below the offset error due to noise.

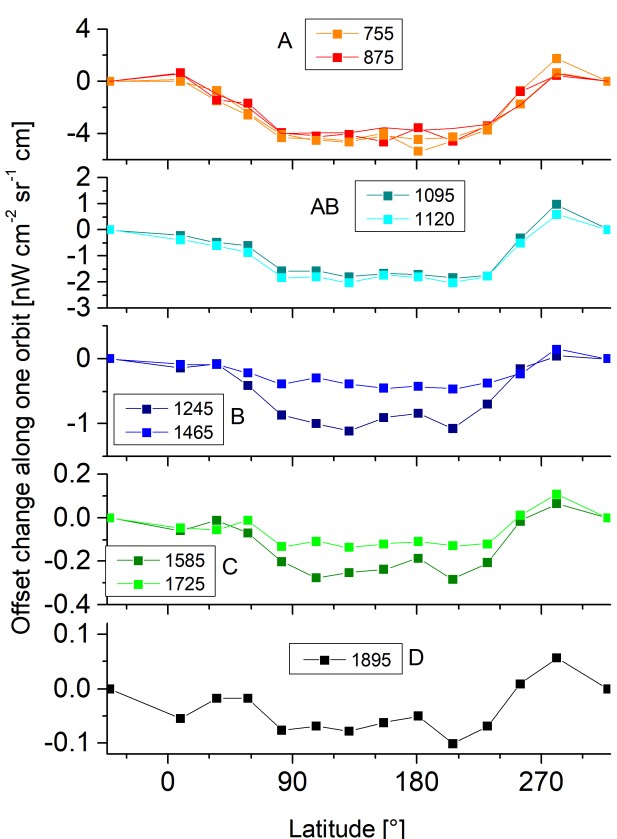

**Figure 9.** Variation of the instrument offset along the orbit for selected wavenumbers in November 2003 (FR mode).



The offset error due to variations in the instrument temperature is spectrally correlated over all bands, it is correlated in time, but not constant, between two offset measurements, and it is strongly vertically correlated although not constant, because the different altitudes are measured at different times and thus at different instrument temperatures.

## 5.6   Uncertainty of the non-linearity correction

Initially, it was planned to monitor the non-linearity by dedicated characterization measurements in flight where the on-board calibration blackbody temperature was varied (so-called IF4 measurements). Unfortunately, the achievable temperature range was too small for a reliable characterization. Therefore, the parameters from the on-ground characterization have been applied to the data of the whole mission up to data version 5. In order to reveal possible changes in the non-linearity over the mission and to improve the non-linearity characterization, an alternative characterization method, the so-called DC-zero method, has been developed using out-of-band artifacts caused by the detector non-linearity (Birk and Wagner, 2010; Kleinert et al., 2015). The out-of-band data are usually suppressed by the on-board filtering and decimation. They are only available in a special raw data mode (so-called IF16 measurements) where the filtering and decimation is switched off. 30 IF16 measurements were acquired throughout the mission, mostly combined with decontamination events. These measurements cover blackbody, deep space and atmosphere.

Using these measurements, it is possible to determine the detector response curve (output as a function of incident photon flux) and to derive the required scaling factors for the non-linearity correction dependent on the interferogram peak-to-peak-value $\mathrm{ADC}_{max-min}$. It turned out that the detector curve changes over time due to detector aging, furthermore it is dependent on instrument temperature and the degree of ice contamination. Therefore, instrument temperature, ice contamination load, and orbit number (i.e. time) serve as further input to calculate the appropriate detector curve.

The DC-zero method utilizes the fact that the DC zero point for all interferograms in the linear domain is the same for 100 % modulation efficiency (Birk and Wagner, 2010). When the modulation efficiency is known, the non-linearity information can be derived from the out-of-band artifacts utilizing scene and calibration IF16 spectra with different integral radiance. The method was tested for the Bruker IFS 125 HR spectrometer at DLR where DC values are available. The agreement of both methods (with and without using the DC values) is within the uncertainty. In principle, the modulation efficiency can be obtained by taking into account the IF4 blackbody measurements, but it turned out that especially for channels with less non-linearity (B1, B2) the derived modulation efficiency results were not reliable. Therefore the modulation efficiency is estimated from the optical specifications and instrument properties to be 91 % (Kleinert et al., 2015). It is assumed that the instrument is well aligned and the modulation efficiency is rather wavenumber independent in the relevant spectral range of 685 to $1500\,\mathrm{cm}^{-1}$.

A multi-dimensional regression in orbit number (equivalent to time), temperature and ice has been applied to the data. The main uncertainties of the determination method are the assumption, that the detector curve is characterized by a 2nd or 3rd order polynomial for the B and A bands, respectively, the estimate of the modulation efficiency, and the regression error. The uncertainty of the resulting scaling factors is estimated to be better than 2 % (Birk and Wagner, 2010).

Since the non-linearity correction is applied to blackbody, deep space and atmospheric measurements, this error leads to both a multiplicative and an additive error in the calibrated spectra. The multiplicative error can be estimated to less than 2 %,





since the errors in the scaling factors of blackbody, deep space and atmospheric spectra are correlated and partly compensate. This compensation effect is best for large atmospheric radiance levels and thus for low tangent altitudes.

For the offset error, the situation is different. The radiance level of atmospheric measurements of high tangent altitudes is close to the one of the deep space spectrum, leading to similar $\mathrm{ADC}_{max-min}$ values. Therefore the scaling factors applied during non-linearity correction are similar, and a resulting offset error is to a large extent compensated. The offset error increases with increasing radiance level, i.e. towards lower tangent altitudes. It is below $5\,\mathrm{nWcm}^{-2}\mathrm{sr}^{-1}\mathrm{cm}$ in the stratosphere and above and below $10\,\mathrm{nWcm}^{-2}\mathrm{sr}^{-1}\mathrm{cm}$ in the troposphere in band A. In band AB, it is below 1 and $2\,\mathrm{nWcm}^{-2}\mathrm{sr}^{-1}\mathrm{cm}$, respectively, and in band B it is below $0.5\,\mathrm{nWcm}^{-2}\mathrm{sr}^{-1}\mathrm{cm}$ and therewith well below the NESR level.

A further error source due to non-linearity is the impact of the cubic artifact on the spectra. The non-linearity not only leads to a different (mean) response depending on the incident photon flux, which is corrected by the appropriate scaling of the interferograms, but it also leads to a distortion of the interferogram peak, leading to artifacts in the spectrum. Quadratic terms of the non-linearity curve lead to out-of-band artifacts and do not distort the signal of interest, whereas cubic terms lead to artifacts inside the nominal spectral range. These artifacts act as an additive contribution to the uncalibrated spectra and with this, they alter the gain function, leading to a scaling error in the calibrated spectrum. The cubic artifact in the atmospheric spectrum leads to an offset error. Both scaling and offset errors are spectrally varying. The left plot of Fig. 10 shows the estimated offset error due to the cubic artifact for channel A2 at a tangent altitude of 52 and 15 km, respectively. The error for A1 is smaller, due to the smaller spectral range (see Fig. 1) and thus the smaller photon load. The offset error is well below the NESR level. The estimated gain error for A1 and A2 is shown in Fig. 10 in the right plot. It is largest for small wavenumbers and is up to $1.8\,\%$ for channel A2 at the beginning of the mission. Due to the detector aging, the error decreases over time. Since the two channels A1 and A2 are combined to one spectral band A, the error in the level 1b data is between the error of A1 and A2. It is estimated to about $1.5\,\%$ at $685\,\mathrm{cm}^{-1}$ and to less than $1\,\%$ above $700\,\mathrm{cm}^{-1}$. For the channels B1 and B2, cubic artifacts are negligible.

The analysis of in-flight measurements with varying blackbody temperature (IF4 measurements) revealed also a small non-linearity for band C. The blackbody measurements taken at different temperatures have been radiometrically calibrated and compared to the expected Planck function. While the values are within $0.1\,\%$ for band D, they show deviations of up to $0.4\,\%$ for band C. An error in the blackbody temperature would have a larger effect on band D than on band C, therefore the deviation in band C is attributed to a small non-linearity effect.

Since the first order effect of the non-linearity error is a scaling error of the uncalibrated spectra, the error is rather wavenumber independent within one band. The error may vary from one band to another, because each detector is characterized independently. Only the neglect of the cubic artifact in band A has a spectral dependency as illustrated in Fig. 10. The error is altitude dependent: The offset error is larger for low tangent altitudes while the gain error is larger for high tangent altitudes. The error also varies in time, since the detector properties change over time and the relation between total photon load and $\mathrm{ADC}_{max-min}$ is also not constant under all circumstances. These variations are not well captured by the sparse characterization measurements. Furthermore, most of the IF16 measurements were taken while the satellite was close to the Kiruna ground station to enable fast enough downlink speed for the raw data mode. Also, these measurements were mostly shortly





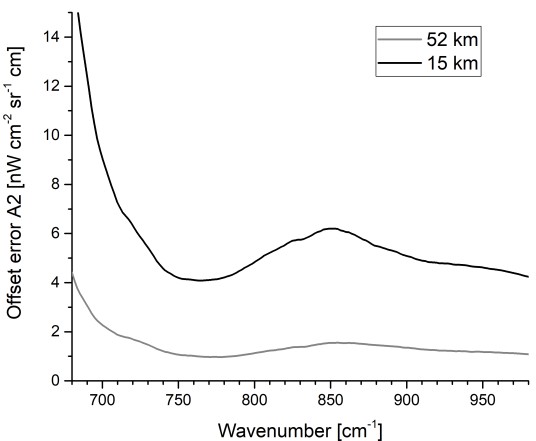
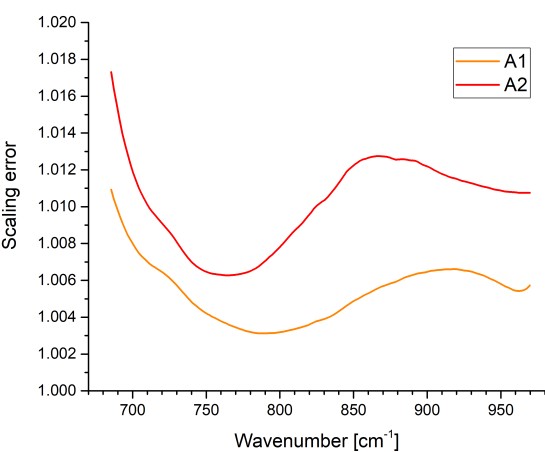

**Figure 10.** Left: Estimated offset error for A2 due to the neglect of the cubic artifact for a tangent altitude of 15 km (black) and 52 km (gray), respectively. Right: Estimated gain error due to the neglect of the cubic artifact in the blackbody and deep space spectra for channel A1 (orange) and A2 (red) for orbit 1680 at the beginning of the mission.

before and after the passive decontamination. Before decontamination the ice load on the detectors was at the maximum while after decontamination the thermal equilibrium may not have been fully established. Thus, the characterization measurements may not be fully representative for the standard measurement situation. The time scales of the non-linearity error can only be estimated from the underlying physical effects, namely detector aging, ice contamination and temperature variations. These

5 effects vary on a time scale of weeks (ice, temperature) to years (aging).

### 5.7 Microvibrations

Microvibrations (introduced by the satellite bus and detector mechanical cooler) are introducing phase modulations in the interferometer. For individual spectra an offset error close to the low wavenumber boundary occurs with up to 1 % of the unperturbed spectral intensity. The error is periodically changing from spectrum to spectrum. Since many spectra are co-added

for the gains, microvibrations are canceled out in the gains but are present in the scene spectra. The expected ghost lines are well below the NESR and thus not detectable in the calibrated spectra. Since the phase of the ghost lines is changing from spectrum to spectrum, they cancel out when co-adding several spectra, e.g. for monthly means.

### 5.8 Pointing jitter

Pointing jitter can be observed in raw data IF16 measurements. Pointing jitter leads to an amplitude modulation of the inter-

15 ferogram, which is strongest in presence of strong atmospheric gradients. The frequency of the pointing jitter is 135 Hz, the amplitude is in the order of 100 m for most of the mission with amplitudes up to 250 m between 2003 and mid 2005. Pointing



jitter can cause ghost lines in the spectra and leads to a small widening of the effective field of view. As for the microvibrations, the phase of the pointing jitter varies from interferogram to interferogram, such that possible ghost lines cancel out when averaging over a larger dataset. Simulations have shown that the expected ghost signatures are within the $1\,\sigma$ NESR levels and thus not easy to detect in calibrated spectra. From retrieval results no obvious impacts related to pointing jitter were found.

## 6 Estimate of the radiometric error from calibrated spectra

In the previous section, the radiometric error has been estimated based on the analysis of the underlying physical effects. In this section, the radiometric error is estimated directly from calibrated spectra. The gain error can be estimated from the comparison of calibrated spectra of different channels in overlapping regions. The offset error is estimated from spectral regions where no atmospheric signal is expected. This quality of this error estimation is limited. Especially the comparison of spectra of different channels in overlapping regions cannot give an absolute error, but it is a good consistency check. Any differences found should be within the error estimated in the previous section.

### 6.1 Estimate of gain error

As shown in Fig. 1, the spectral channels of the different regions show a certain overlap before digital filtering, decimation and channel combination. Since these steps are usually already performed on-board, the overlapping regions are only available in IF16 measurements, where the raw interferograms are directly sent to ground. When calibrating these measurements, it is possible to deduce a scaling error by determining the correlation between the data from different channels: The radiances of one channel are plotted vs. the corresponding radiances of the other channel. A straight line is fitted to this scatter plot, resulting in slope and offset which should ideally be 1 and 0, respectively. The deviations from the ideal values are used for error assessment. The slope was determined for all overlapping channels and all available IF16 orbits using all available scene spectra. Differences between channels point towards a radiometric error in at least one of the channels. This method does not allow for an absolute error quantification, but it is a valuable check of the self-consistency of the data.

Unfortunately, the number of IF16 measurements over the mission is sparse (only 30), and the number of altitude scans is limited (1 to 4 per orbit). Scaling ratios have been determined for each available sweep using the following overlapping spectral ranges (all numbers in $\mathrm{cm}^{-1}$):

| | |
|---|---|
| A2 / A1 | 700 - 800 |
| B1 / A2 | 1000 - 1070 |
| B2 / B1 | 1200 - 1500 |
| C2 / C1 | 1550 - 1750 |
| D2 / D1 | 1850 - 2400 |

The values show a large scatter, but no systematic forward-reverse differences have been found, and the altitude dependency is rather small. Therefore, the median value for each orbit has been used as indicator for a scaling difference between overlapping channels. The median instead of the mean has been chosen in order to be more resistant to outliers. The results are shown in Fig. 11. A linear fit to the data has been added in order to reveal a possible trend. The data for B2/A1 has been



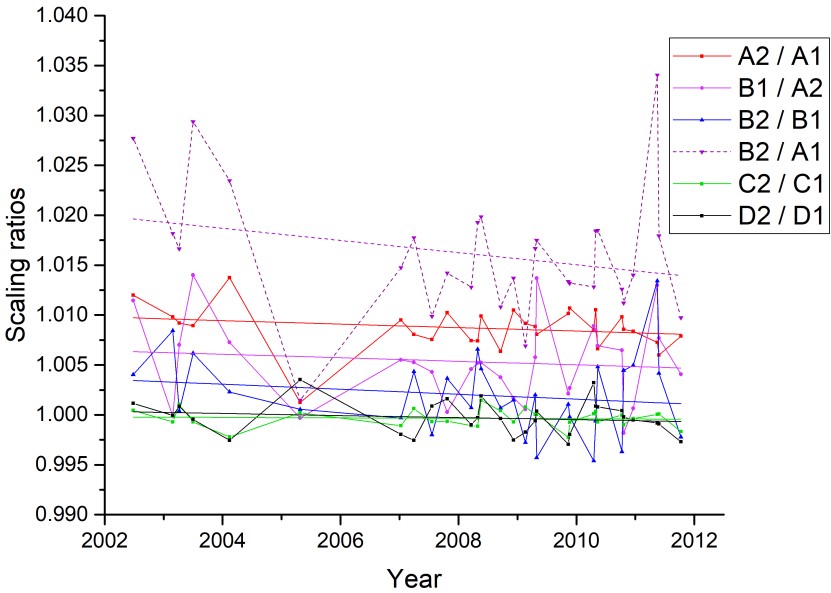

**Figure 11.** Scaling ratios between overlapping channels deduced from IF16 measurements over the mission. A linear fit to the data has been added.

calculated from the ratios B2/B1, B1/A2, and A2/A1. While the ratios for the linear channels C and D are very close to 1, the non-linear channels show systematic differences up to 1 %. Since these differences are all positive, they add up to an inconsistency between channel A1 and channel B2 of about 2 % at the beginning of the mission. The linear fit shows a small trend towards smaller differences at the end of the mission. The values for the individual orbits, however, show a rather large scatter

of sometimes more than 1 %. Overall, the differences found between the different channels can be explained with the estimated errors for the temporal gain variation (5.2) and the non-linearity correction (5.6).

The consistency between the channels A1 and A2 can also be deduced from nominal data. Because of the non-linearity correction, which is different for A1 and A2 and which is performed on ground, the combination of channels A1 and A2 to band A is also performed on ground. It is thus possible to process A1 and A2 separately and to compare the results. This has

been done for 14 orbits throughout the mission, 2 in FR mode and 12 in OR mode. Other than for the IF16 measurements, where only 1 to 4 scans per orbit were available, the data in nominal mode provides data over the full orbit. This allowed to calculate a mean scaling difference over the orbit for each of the 27 tangent altitude levels. It was not possible to determine a scaling difference for the uppermost tangent altitude, because the atmospheric signal is too weak. In FR mode, the altitude range was covered by only 17 instead of 27 tangent altitudes, therefore the FR data has been interpolated to 27 altitude levels

to allow for a better comparison. The scaling difference is shown in Fig. 12. The agreement is mostly within 0.5 to 1.5 %, well in line with the ratios deduced from the IF16 measurements. The differences are generally larger for higher altitudes, which



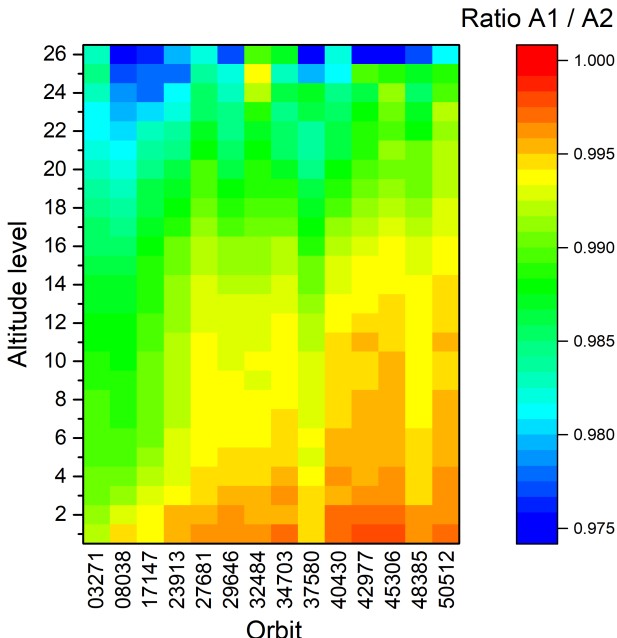

**Figure 12.** Ratio of calibrated spectra of channel A1 and A2.

points towards an error in the non-linearity correction and rules out other error sources like a slightly different field of view. In this case, relative differences should be larger for lower tangent altitudes where the gradient of the atmospheric signal is much stronger. The difference slightly decreases towards the end of the mission, which is also in line with the IF16 data. There is, however, a certain variation in time, e.g. the difference for orbit 37580 is larger than for the neighboring orbits.

5 ## 6.2 Estimate of offset error

The offset error can be estimated directly from calibrated spectra from spectral regions where no atmospheric signal is expected. This works especially well for high tangent altitudes, but in band A the offset can be determined down to about $30\,\mathrm{km}$ in the atmospheric window. Above $65\,\mathrm{km}$, mean radiances of the uppermost tangent altitude of different measurement modes have been calculated for selected spectral intervals where no atmospheric signal is expected. In order to reduce the noise level, 10 orbital mean values have been calculated in the following spectral regions (all numbers in $\mathrm{cm}^{-1}$):

| A | 840 - 870 |
| AB | 1140 - 1170 |
| B | 1215 - 1235 |
| C | 1724 - 1729 |
| D | 1985 - 2015 |

In these spectral regions, the atmospheric contribution is estimated to be below $0.05\,\mathrm{nW cm}^{-2}\mathrm{sr}^{-1}\mathrm{cm}$ above $60\,\mathrm{km}$ from forward calculations. The offset, i.e. the mean spectral radiance, has been calculated for the uppermost tangent altitude in



different measurement modes: Nominal mode (NOM, about 70 km), middle atmosphere mode (MA, about 100 km), and upper atmosphere mode (UA, about 170 km). The data used (226185 spectra in total) have been separated in FR and OR mode, furthermore they have been analyzed separately for day and night and for forward and reverse sweep direction. The offset values are summarized in Tab. 2, together with the $1\sigma$ standard deviation and the NESR for comparison.

**Table 2.** Offset values determined from calibrated spectra. All values are in $\mathrm{nWcm^{-2}sr^{-1}cm}$

| altitude | | A | AB | B | C | D |
|---|---|---|---|---|---|---|
| 70 km | offset | 2.45 | 0.96 | 0.58 | 0.09 | 0.15 |
| | day/night difference | 0.68 | 0.28 | 0.22 | 0.06 | 0.11 |
| | f/r difference (FR) | 1.57 | 0.89 | 0.63 | 0.14 | 0.05 |
| 100 km | offset | 1.99 | 0.66 | 0.47 | 0.06 | 0.03 |
| | day/night difference | 0.59 | 0.23 | 0.25 | 0.04 | 0.01 |
| | f/r difference (FR) | 1.64 | 1.13 | 0.55 | 0.13 | 0.07 |
| 170 km | offset | 1.00 | 0.33 | 0.24 | 0.03 | 0.01 |
| | day/night difference | 0.59 | 0.26 | 0.21 | 0.04 | 0.02 |
| | standard deviation | 1.6 | 1.2 | 0.8 | 0.3 | 0.12 |
| | NESR (OR) | 17.1 | 9.9 | 7.9 | 3.2 | 1.0 |

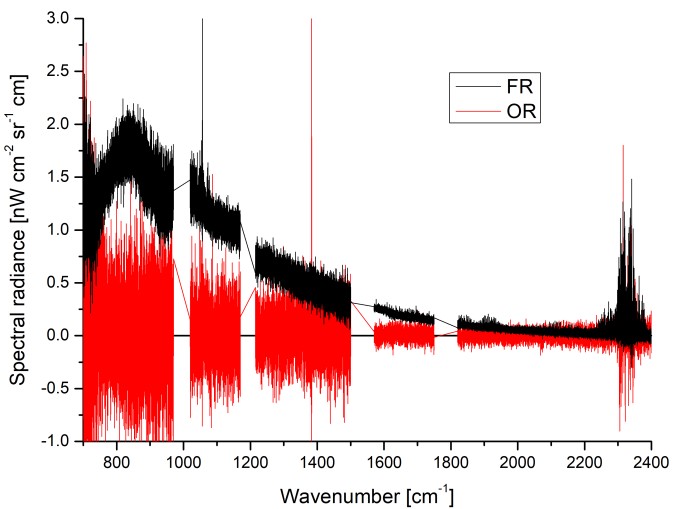

**Figure 13.** Difference between spectra of forward and reverse sweep direction for FR mode (red) and OR mode (blue) at about 70 km tangent altitude. 35000 and 3800 spectra have been co-added per sweep direction for FR and OR mode, respectively.





There is a systematic positive offset in the data, which has also been observed by López-Puertas et al. (2009) and Günther et al. (2018). The offset is decreasing with altitude and wavenumber. The data also reveals a systematic day-night difference with higher values at daytime. Furthermore, a systematic forward-reverse difference can be observed in full resolution mode. This difference disappears in optimized resolution mode (see Fig. 13). The offset is about one order of magnitude below the

NESR and is therefore not visible in single spectra.

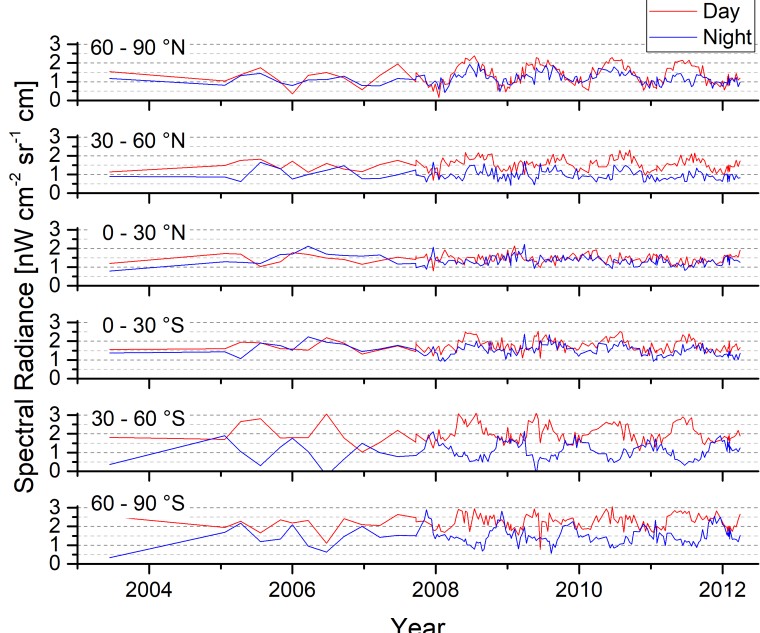

**Figure 14.** Mean offset over 100 to 170 km tangent altitude in band A, separated in 6 latitude bands and separate for day (red) and night (blue).

In order to reveal offset variations over time and/or latitude, spectra from upper atmosphere measurements in the range of 100 to 170 km tangent altitude have been analyzed for six latitude bands (see Fig. 14). For each latitude band, measurements of typically one day of the whole altitude range have been co-added, separate for day and night. This leads to about 1000 co-added spectra per data point. The result is shown in Fig. 14 for band A. Upper atmosphere measurements were rather sparse at the

beginning of the mission but were regularly acquired about every 10 days from November 2007 onwards. The figure shows a seasonal variation of about $1.5\,\mathrm{nW cm^{-2} sr^{-1} cm}$ at high latitudes. At southern latitudes, this variation is anticorrelated between day and night while it is correlated at northern latitudes. Depending on the season, there is a latitudinal variation of the offset of up to $2\,\mathrm{nW cm^{-2} sr^{-1} cm}$. The variation of the offset is similar in the other bands, with smaller amplitude, corresponding to the generally smaller offset. The latitudinal variation of the offset is similar for the whole altitude range investigated (M.

López-Puertas, personal communication, 2008).





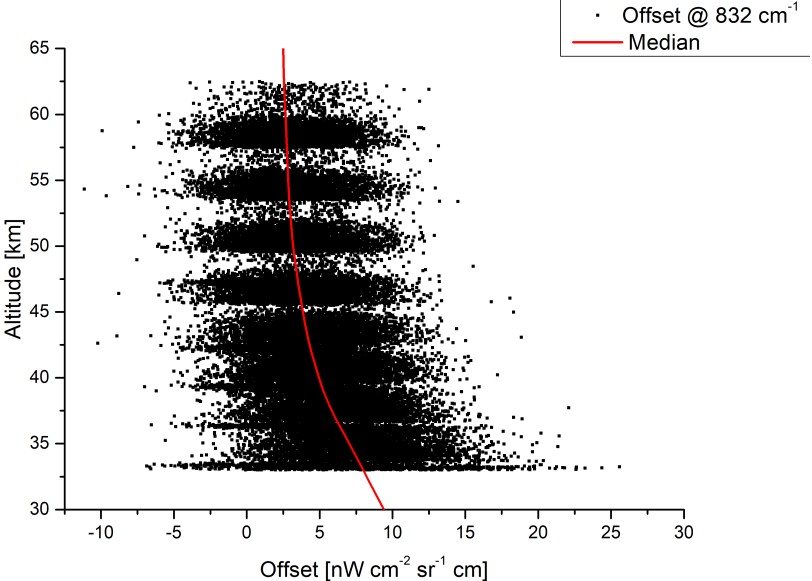

**Figure 15.** Offset determined from calibrated spectra in an altitude range of 33 to 63 km around $832\,\mathrm{cm}^{-1}$.

In band A, the offset in calibrated spectra can also be estimated for lower tangent altitudes, because no broadband atmospheric contribution is expected around $832\,\mathrm{cm}^{-1}$ down to about 33 km. The offset in the altitude range between 33 and 63 km has been determined from about 4000 orbits throughout the mission from a joint retrieval of $H_2O, O_3, NO_2, F11$, F22 and offset ((M. Höpfner, personal communication, 2016). The result is shown in Fig. 15. Despite a large scatter of the values, a

5 mean positive bias is obvious. At high altitudes, the offset is around $2.5\,\mathrm{nWcm}^{-2}\mathrm{sr}^{-1}\mathrm{cm}$ in line with the values found for the uppermost tangent altitude in nominal mode. When going further down, the offset is systematically increasing. At 33 km, the offset is about $8\,\mathrm{nWcm}^{-2}\mathrm{sr}^{-1}\mathrm{cm}$. Since an increasing offset with decreasing tangent altitudes has been observed in all spectral bands between 150 and 68 km, it is expected that the increase below 68 km is also similar in all bands. Therefore, the offset error at 33 km is estimated to be 3.1, 1.9, 0.3, and $0.15\,\mathrm{nWcm}^{-2}\mathrm{sr}^{-1}\mathrm{cm}$ in bands AB, B, C, and D, respectively.

The forward-reverse difference can be attributed to a calibration error. It is only present during the FR part of the mission, and it is constant over time and independent of tangent altitude. This error cancels out when averaging over time because of the odd number of sweeps in one limb scan. The data is automatically averaged over forward and reverse measurements. The offset variation with altitude cannot be completely explained with instrument effects. Part of this offset could be related to the cubic non-linearity artifact (see Fig. 10, left), but the offset error introduced by this artifact is too small to explain the whole

offset observed. Therefore it is assumed that there is a certain straylight contribution from earth or clouds.



# 7 Spectral accuracy

The spectral axis is scaled according to the wavelength of the reference laser. The spectral calibration factor (SCF) is determined on a daily basis, and the SCF is updated together with the gain function. Figure 16 shows in red the variation of the SCF over the mission as determined by the spectral calibration. The variation from one SCF determination to the next is depicted

5  in blue on the right axis. It is dominated by the noise of the SCF determination. This variation is used as estimate for the spectral calibration accuracy. The accuracy is mostly within 0.14 ppm in the FR period and within 0.27 ppm in the OR period, corresponding to a spectral shift of $0.0004\,\mathrm{cm}^{-1}$ and $0.00065\,\mathrm{cm}^{-1}$, respectively, at $2410\,\mathrm{cm}^{-1}$. This is well within the requirement of $0.001\,\mathrm{cm}^{-1}$.

This error increases linearly with wavenumber, it is fully vertically correlated and it is fully correlated in time until a new

10  SCF is applied (usually one day).

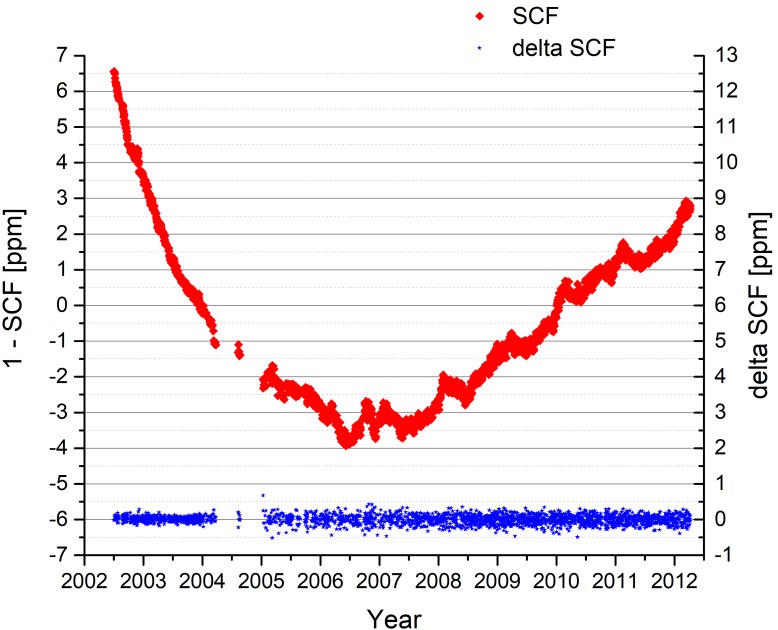

**Figure 16.** Spectral calibration factor (SCF) as determined from atmospheric measurements over the mission (red). The difference between subsequent SCF-values is depicted in blue on the right axis.

# 8 Line of sight accuracy

Achieving a good LOS accuracy at tangent point for a limb sounder is very challenging. E.g., in rearward an error of 0.01 deg on the pointing angle corresponds to 0.5 km at the tangent point. Dedicated LOS calibration measurements have been acquired




in a mode where the instrument is pointed at stars on a weekly basis. The pointing errors were calculated from the expected and actual time of the star passing through the IFOV. Figure 17 presents the pointing errors determined along the mission. At the beginning of the mission, the random variation corresponds to an on-board satellite attitude control software bug which was corrected in December 2003. Toward the end of mission, the calibration was no longer possible due to a detector noise increase.

From the data, an annual cycle and negative trend has been deduced. This behavior was also observed with other instruments on-board the satellite along with a validation campaign of MIPAS retrieved ozone against ozone measured at ground stations (Hubert et al., 2016, p. 36). A model has been fitted to the data and is used to correct the altitude in level 1b processor version 8.

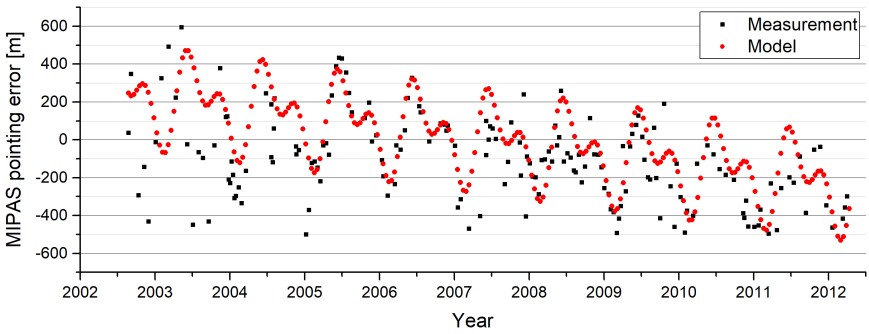

**Figure 17.** MIPAS pointing errors along the mission (grey) and fitted error model (red).

The engineering tangent altitudes reported in the level 1b product have been validated against an independent temperature and LOS retrieval (von Clarmann et al., 2003). For the version 8 data, the retrieved tangent altitudes are generally higher than

the engineering tangent altitudes. The overall offset is in the order of 0 to 400 m over the mission. At low tangent altitudes, differences of up to 700 m have been observed with typical differences of 300 to 500 m (M. Kiefer, personal communication, 2017). The higher error in the troposphere is related to atmospheric refraction. The level 1b processor uses a standard atmosphere in the calculation, and the difference between the actual atmosphere state and the standard model leads to an additional error. The overall error is well below the requirement of $\pm\,1800$ m. The accuracy of the latitude and longitude is estimated to

$\pm\,0.021\,^\circ$ and $\pm\,0.004\,^\circ$, respectively.

## 9   Summary of the level 1b data accuracy

The various sources of uncertainties are summarized in Tab. 3. The different sources for scaling and additive error are summed up quadratically to give an overall scaling and additive error estimate. For each error source, the spectral, spatial (vertical) and temporal correlation is characterized. In some cases, two values are given: a typical value and an upper limit in brackets. This

upper limit refers to either only a small spectral range of the band or short time periods during the mission. Details about the individual errors are given in the respective sections above.



**Table 3.** Summary of the level 1b data accuracy. For details, see text.

| | | | Spectral band | | | | | | Correlation | | |
|---|---|---|---|---|---|---|---|---|---|---|---|
| | | | A | AB | B | C | D | | spectral | altitude | time |
| **NESR** | | FR | 30 (80) | 16 | 16 | 3 | 3 (5) | | - | - | - |
| (nW cm$^{-2}$ sr$^{-1}$ cm) | | OR | 20 (50) | 10 | 10 | 2 | 2 (3) | | | | |
| **Scaling accuracy** | Gain noise | | 0.1 | 0.1 | 0.1 | 0.1 | 0.4 (1.2) | | resol. (a) | full | 1 day |
| (%) | Gain variation | | 0.4 (1.5) | 0.3 (1.5) | 0.4 (2) | 0.4 (1.2) | 0.6 (1.2) | | band | full | 1 day |
| | Blackbody | | 0.5 | 0.5 | 0.5 | 0.5 | 0.5 | | high (b) | full | 1 day |
| | NL determination | | 2 | 2 | 2 | 0.4 | - | | band | full (c) | weeks to years |
| | Cubic artifact | | 1 (1.5) | 0.1 | 0.1 | - | - | | band (d) | full (c) | mission (e) |
| | *Total* | | *2.4* | *2.1* | *2.1* | *0.8* | *0.9* | | | | |
| **Offset accuracy** | Offset noise | FR | 6 (20) | 3 | 2.5 | 0.7 | 0.6 (1) | | resol. (a) | full | 300s / 700s |
| (nW cm$^{-2}$ sr$^{-1}$ cm) | | OR | 3 (10) | 2 | 1.5 | 0.4 | 0.15 (0.3) | | | | |
| | Offset drift | FR | 2 | 1 | 0.5 | 0.1 | 0.05 | | full | full | 300s / 700s |
| | | OR | 4 | 2 | 1 | 0.2 | 0.1 | | | | |
| | NL determination | | 5 (10) | 1 (2) | 0.5 | - | - | | band | full (f) | weeks to years |
| | Cubic artifact | | 5 (10) | - | - | - | - | | band (d) | full (f) | mission (e) |
| | *Total* | | *9.5* | *3.3* | *2.6* | *0.7* | *0.6* | | | | |
| **Spectral accuracy** | | FR | 0.14 | | | | | | full | full | 1 day |
| (ppm) | | OR | 0.27 | | | | | | | | |
| **LOS (m)** | | | 400 (700) | | | | | | full | full (f) | not known |

Comments:    (a) according to the spectral resolution of the calibration measurements
                    (b) depending on spectral emissivity
                    (c) increasing with altitude
                    (d) highly correlated but not constant within one band
                    (e) decreasing with time
                    (f) decreasing with altitude

## 10  Conclusions

We have quantified the MIPAS level 1b error in terms of radiometric, spectral, and line of sight accuracy. The thorough characterization of the instrument and level 1b processing has led to several improvements in the latest level 1b processing version 8 compared to earlier processing versions. The radiometric error has been separated into a multiplicative gain error and an additive offset error, and the different types of error have been characterized in terms of spectral and vertical correlation lengths and in terms of evolution in time. The error correlation is important for its impact on the retrieved species, e.g., errors with short correlation lengths in time cancel out when averaging over a longer time span.

The estimated accuracy has been cross-checked by analyzing the self-consistency of calibrated spectra. From special measurements, it could be shown that scaling differences between the data acquired by different detectors are within the estimated





gain errors. The offset error is deduced from calibrated spectra using spectral regions and altitude ranges where no atmospheric signal is expected. At high tangent altitudes, this error is rather below the error estimated from the characterization, but it increases systematically with decreasing altitude, which is not expected from instrument characterization. Therefore it is assumed that this effect is related to stray light rather than an instrumental offset.

The errors are well within specifications, and the achieved accuracy allows to retrieve atmospheric parameters from the measurements with high quality. It should be noted, however, that the analysis of trends is very sensitive to long-term drifts of instrument properties, namely changes in the non-linearity of the photoconductive detectors.

The experience with the MIPAS instrument has shown that thorough characterization work is extremely important for a good data quality throughout the mission. Regular characterization measurements are indispensable in order to reveal instrument

changes, e.g. due to aging, and the regular transmission of raw, unprocessed data is very valuable to understand the instrument and identify possible issues. Flexibility must be allowed in operation mode and the calibration process to cope with changing situations in long term missions. Last but not least an exhaustive on-ground characterization of parameters which cannot be determined during flight is very valuable for understanding the data measured in flight and also improves the data quality. These aspects should be considered for any future satellite mission.

*Competing interests.* The authors declare that they have no conflict of interest.

*Acknowledgements.* This work has been performed under ESA contract no 4000112275/14/I-LG. The support in data processing and analysis by Ginette Aubertin is greatly acknowledged. We thank the members of the MIPAS Quality Working Group for many fruitful discussions and valuable data analyses.

We acknowledge support by the Deutsche Forschungsgemeinschaft and Open Access Publishing Fund of Karlsruhe Institute of Technol-

ogy.





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
