# Peer review of "Level 1b error budget for MIPAS on ENVISAT"

_Atmospheric Measurement Techniques, 2018_

## Referee Comment (RC1) · Anonymous Referee #1 · 25 Jul 2018

I find the work described in this paper complete and detailed enough to be of a great usefulness for the MIPAS data users community.

The presentation is very clear, but I find that in some figures the choice of colors (some of which are really similar), line styles and symbols do not help readability, specially in the most "crowded" plots in which several lines are plotted together.

A technical remark: in section 2 MIPAS is described as a Michelson interferometer, but I'd define it more of a Mach-Zehnder design.

At the end of section 2 I'd use "trade-off" instead of "trade".

In section 3 the authors refer to a "simple correction algorithm" for the spike correction: it would be desirable, if possible, to have more detail, or a literature reference.

Page 13, line 30: the phrasing is not clear, possibly the comma after "assumption"

[Figure]

needs to be removed.

Figure 13: it seems that the colors in the caption are not in agreement with the ones used in the figure.

Last, a more general remark about section 6.2 (and the paper in general): there is a lot of detail in the characterization and description of the measured contributions to the error, but in several places I find that some kind of physical interpretation of the measured phenomena (e.g. seasonal cycles) could be provided together with the characterization. In section 6.2 there is a couple of sentences providing an interpretation for some of the measured effects, but I think the work could gain a lot of significance if this could be expanded and applied in each case where a significant and characteristic effect is detected.

---

## Referee Comment (RC2) · Anonymous Referee #2 · 6 Aug 2018

The article "Level 1b error budget for MIPAS on ENVISAT" by Anne Kleinert et al. gives a comprehensive overview on the error budget of the level 1b data in the upcoming and probably last version 8. The level 1b data are the calibrated and geo-located spectra that serve as the basis for the retrievals of the atmospheric state and constituents. The error budget presented here is of great value, especially to all who are retrieving any species from MIPAS data and all who are using derived MIPAS products. The paper is well written and structured. I recommend it for publication in AMT after some minor and technical corrections.

**Minor comments**
page(p)1 line(l)18: The examples for clouds comprise only PSCs and PMCs. I'd suggest to also add an example for upper troposphere/ lower stratosphere cirrus clouds e.g. Spang et al., ACP, 2012 or Sembhi et al., ACP, 2012

p2 l17−18: Please consider adding a sentence for explanation here. It takes some time studying Fig. 1 to understand how 4 detectors can cover 5 spectral bands even if one detector fails.

p4 l5: Please add some description of the parameters given in Table 1. E.g., what does coadditions per gain measurement mean?

p4 l14: What does simple mean here? Is the correction algorithm cutting the spikes?

p6 l8: Please state what low means. 10 km, 20 km or 50 km? Is this factor linear from top to bottom?

p6 l8−10: Does this have any practical implications? Does this mean that when investigating measurements at e.g 10 km all data points with radiance $\leq$ NESR$\times$1.5 should be discarded / are not significant? Please clarify.

p7 l14: Please add to which value the requirement of the scaling accuracy was relaxed.

p12 l4−8 & Figure 9: There are 3 spectral bands shown in Figure 9 for band A (best visible at 180° Latitude), but the label and text indicate only 2 bands. Further, what does Latitude$<$0, $>$90, $<$180, $<$270 mean? Where is the Equator? Is each data point representing a single sweep or a scan measurement? In the text it says that in FR mode the variation between two subsequent measurements is below 2 nWcm$^{-2}$sr$^{-1}$cm in band A, but in Figure 9 it goes down to $-4$ nWcm$^{-2}$sr$^{-1}$cm. Please clarify.

p13 l26−28: Did you assume 91% modulation efficiency for all bands? Or did you derive it using the DC-zero method for all detectors but B1 and B2? Please clarify.

p13 l31: And what about bands AB, C, D?

p18 Figure 12: Is my assumption correct that altitude level 1 is low (about 10 km) and altitude level 26 is high (about 65 km) altitude? Please provide a description in the caption. Please consider a change of the color scale to blue/green for excellent agreement (ratio close to 1) and orange/red for slightly reduced agreement (ratio smaller than 1).

p20 l1 − 3, l6-15: Do you have any idea what is causing the day-night difference, latitudinal and seasonal variation of the offset?

p22 l6: Please explain why the spectral accuracy is given like a mixing ratio in ppm. I'd rather expected values in cm$^{-1}$ as you provide in line 7.

p23 l12: Which standard atmosphere did you use (the U.S. Standard Atmosphere or any other)? Is this atmosphere anywhere available?

**Technical comments**
Please revise all places where you are using "allow to".
p1 l9: please write "... allows atmospheric parameters to be retrieved ..."

Text font and unit font seem to be different throughout the manuscript e.g p1 l13.

p1 l15-16: please write "...measurements allow for retrievals of..."

p1 l23: it should be $SF_6$ and $CF_4$

p2 l13: See first technical comment. Please write e.g.: "It allows two-sided ... up to $\pm 20$ cm to be measured."

p2 l15: cooled "to" 70 K

p2 l18: Please write "bands".

p4 l20: "... consist of scaling ..."

p5 l8: Please write: "The level 1b processor reports the geolocation with each measured spectrum..."

p7 l11: Please consider adding the channel names in brackets here to facilitate reading. E.g. "... the non-linear channels (A, AB, B)..." and "linear channels (B,C)..."

p11 l6: See first technical comment. Please write e.g. "This allows us to conclude...".

p13 l2: Please check comma placement between "constant" and "between"

p17 l6: Please add "Section" before "5.2" and "5.6".

p17 l11: See first technical comment. Please write e.g. "This allowed for calculating a mean..."

p18 l1: Please replace "like" with "such as"

p19 Figure 13: Please fix the caption. The picture label says that the FR curve is black and the OR curve is red.

p20 l2: Please consider writing "The offset is decreasing with increasing altitude and wavenumber." for better comprehensibility.

p22 l5: "... used as an estimate for..."

p22 l10: Did you mean "one per day"?

p24 l6: Please add comma after "lengths".

p25 l5: See first technical comment. Please write e.g. " ... allows retrieving atmospheric..."

p25 l8: "...has shown that a thorough..."

p25 l12: Please add comma before "which".

**References**

Spang, R., Arndt, K., Dudhia, A., Höpfner, M., Hoffmann, L., Hurley, J., Grainger, R. G., Griessbach, S., Poulsen, C., Remedios, J. J., Riese, M., Sembhi, H., Siddans, R., Waterfall, A., and Zehner, C.: Fast cloud parameter retrievals of MIPAS/Envisat, Atmos. Chem. Phys., 12, 7135-7164, https://doi.org/10.5194/acp-12-7135-2012, 2012.

Sembhi, H., Remedios, J., Trent, T., Moore, D. P., Spang, R., Massie, S., and Vernier, J.-P.: MIPAS detection of cloud and aerosol particle occurrence in the UTLS

with comparison to HIRDLS and CALIOP, Atmos. Meas. Tech., 5, 2537-2553, https://doi.org/10.5194/amt-5-2537-2012, 2012.

---

## Author Comment (AC1) · 12 Sep 2018

Answers to the interactive comment on "Level 1b error budget for MIPAS on ENVISAT" by Anne Kleinert et al. by Anonymous Referee 1.

We thank Anonymous Referee #1 for reviewing the paper and the valuable comments. Our answers to the comments are given below. Relevant referee comments are inserted *in italics*.

*The presentation is very clear, but I find that in some figures the choice of colors (some of which are really similar), line styles and symbols do not help readability, specially in the most "crowded" plots in which several lines are plotted together.*

The idea behind the color selection was to use similar colors for the detector "pairs" (although they should still be discriminable). In order to enhance readability, we will

use cyan and blue for channels B1 and B2, respectively, and grey instead of dark grey for channel D1. In the plots where the five spectral bands are shown, we will change the colors for AB and B accordingly and use grey instead of black for band D.

*A technical remark: in section 2 MIPAS is described as a Michelson interferometer, but I'd define it more of a Mach-Zehnder design.*

Although the design of MIPAS is not identical to the original Michelson interferometer, we think that it is appropriate to describe MIPAS as a Michelson interferometer, rather than a Mach-Zehnder interferometer.

*At the end of section 2 I'd use "trade-off" instead of "trade".*

Ok

*In section 3 the authors refer to a "simple correction algorithm" for the spike correction: it would be desirable, if possible, to have more detail, or a literature reference.*

If a spike is detected, the values of the affected data points are divided by 2 until they are below a threshold defined by the adjacent points not affected by the spike. We will add this information in the text and add a reference to the Algorithm Technical Baseline Document (ATBD) for MIPAS Level 1B Processing for more detailed information.

*Page 13, line 30: the phrasing is not clear, possibly the comma after "assumption" needs to be removed.*

We will re-phrase the sentence by:

"There are three main sources of uncertainty for the determination of the non-linearity: (1) the assumption, that the detector curve is characterized by a 3rd order polynomial for channels A1 and A2 and by a 2nd order polynomial for channels B1 and B2, (2) the estimate of the modulation efficiency, and (3) the regression error."

*Figure 13: it seems that the colors in the caption are not in agreement with the ones used in the figure.*

[Figure]

Yes, sorry. We will correct the caption.

*Last, a more general remark about section 6.2 (and the paper in general): there is a lot of detail in the characterization and description of the measured contributions to the error, but in several places I find that some kind of physical interpretation of the measured phenomena (e.g. seasonal cycles) could be provided together with the characterization. In section 6.2 there is a couple of sentences providing an interpretation for some of the measured effects, but I think the work could gain a lot of significance if this could be expanded and applied in each case where a significant and characteristic effect is detected.*

We agree that the value of the work is increased by giving a physical background of the error sources and explaining the measured phenomina. There are, however, some effects which we cannot explain. Especially for the offset variation described in section 6.2 we have not found any instrument effect that could explain this behavior. Also a correlation with cloud cover, which could give a hint to straylight effects, has been investigated but did not give a clear picture. Therefore we are limited to reporting the observed offset variation. The variation with day/night, season, and altitude shall demonstrate the variability of the offset. We will add an additional sentence at the end of section 6.2 in order to make the limits of this analysis more clear:

"Also the day/night variation as well as the seasonal, latitude dependent variation of the offset cannot be explained with known instrument effects, but the observed offset variation gives an impression of the expected offset error and its variation."

---

## Author Comment (AC2) · 12 Sep 2018

We thank Anonymous Referee #2 for reviewing the paper and the valuable comments. Our answers to the comments are given below. Relevant referee comments are inserted *in italics*.

*page(p)1 line(l)18: The examples for clouds comprise only PSCs and PMCs. I'd suggest to also add an example for upper troposphere/ lower stratosphere cirrus clouds e.g. Spang et al., ACP, 2012 or Sembhi et al., ACP, 2012.*

We will add the publication of Sembhi et al. (2012) as reference.

*p2 l17 – 18: Please consider adding a sentence for explanation here. It takes some time studying Fig. 1 to understand how 4 detectors can cover 5 spectral bands even if one detector fails.*

[Figure]

We will give the information on the channels and bands first and then add an explanation on the spectral coverage of the channels by writing:

"The spectra from the 8 detectors are summarized in 5 spectral band (denoted A, AB, B, C, and D in Fig. 1). The spectral coverage of the individual detectors is different for the two ports (see Fig. 1) in order to ensure full spectral coverage even if one detector fails. Channel A2, which is optimized for the spectral range of band A, also covers the range of band AB, and channel B1, which is optimized for band AB, also covers band B."

*p4 l5: Please add some description of the parameters given in Table 1. E.g., what does coadditions per gain measurement mean?*

We will change "number of coadditions" to "number of spectra co-added"

Furthermore we will spend a few more words on the calibration in the text explaining the parameters given in Table 1:

"For radiometric calibration, the instrument points towards the internal blackbody or into deep space, i.e. at a tangent altitude of about 210 km. The radiometric gain is determined from pairs of blackbody and deep space measurements on a daily basis, and additional deep space measurements are performed for offset determination several times per orbit. In order to enhance the signal-to-noise-ratio, several spectra are co-added for the calibration measurements. Gain and offset are determined individually for the two sweep directions of the interferometer."

*p4 l14: What does simple mean here? Is the correction algorithm cutting the spikes?*

If a spike is detected, the values of the affected data points are divided by 2 until they are below a threshold defined by the adjacent points not affected by the spike. We will add this information in the text and add a reference to the Algorithm Technical Baseline Document (ATBD) for MIPAS Level 1B Processing for more detailed information.

*p6 l8: Please state what low means. 10 km, 20 km or 50 km? Is this factor linear from*

*top to bottom?*

"low" means the lowermost tangent altitude, i.e. below 10 km. We will add this information in the text.

The additional contribution to the NESR is due to the photon noise of the atmospheric signal. Therefore this factor is not linear but dependent on the vertical profile of the atmospheric spectral radiance in the respective spectral range.

*p6 l8 – 10: Does this have any practical implications? Does this mean that when investigating measurements at e.g 10 km all data points with radiance <= NESR×1.5 should be discarded / are not significant? Please clarify.*

This information is given here in order to provide an overview of the expected NESR range. The practical implications depend on the setup for the retrieval of atmospheric parameters. The NESR of each spectrum is part of the level 1b product and is considered in the retrieval process.

We will change the text to:

"For atmospheric measurements, the NESR is larger than the $NESR_0$ because of the increasing photon load on the detectors. The NESR for atmospheric measurements at low tangent altitudes (below 10 km) is about 20 to $50\%$ larger than the $NESR_0$, depending on the strength of the atmospheric signal in the different bands (not shown)."

*p7 l14: Please add to which value the requirement of the scaling accuracy was relaxed.*

We will add: "was relaxed to $5\%$"

*p12 l4 – 8 Figure 9: There are 3 spectral bands shown in Figure 9 for band A (best visible at $180°$ Latitude), but the label and text indicate only 2 bands.*

In band A the data was shown separately for channel A1 and A2. Since the offset drift along the orbit is quite similar, we will change the figure such that the mean of the two channels is shown.
*Further, what does Latitude<0, >90, <180, <270 mean? Where is the Equator?*

$0°$ and $180°$ denote the equator, while $90°$ and $270°$ denote the north and south pole, respectively. We will clarify this in the figure by adding to the abscissa "Eq, $90°$ N, EQ, $90°$ S" at 0, 90, 180, and 270 degrees, respectively.

*Is each data point representing a single sweep or a scan measurement?*

For each offset measurement, 6 interferograms are aqcuired in FR mode (3 forward, 3 reverse, see Table 1). Even when co-adding the 6 resulting spectra, the noise of these offset measurements is too high to reveal the offset variation along the orbit. Since the offset measurements are always taken at the same latitude, the offset spectra of 15 orbits of November 2003 were co-added (i.e. 90 spectra per measurement point).

We will add this information in the text.

*In the text it says that in FR mode the variation between two subsequent measurements is below $2\,\mathrm{nWcm^{-2}sr^{-1}cm}$ in band A, but in Figure 9 it goes down to $-4\,\mathrm{nWcm^{-2}sr^{-1}cm}$.*

The overall variation along the orbit is up to $4\,\mathrm{nWcm^{-2}sr^{-1}cm}$, but the largest variation between two measurement points (e.g. from point #4 to point #5 between 60 and $80°$ N) is around $2\,\mathrm{nWcm^{-2}sr^{-1}cm}$.

*Please clarify.*

We will change the paragraph on the temporal variation of the offset to:

"The instrument self emission varies slightly along the orbit. This is well captured by the regular offset measurements. Figure 9 shows the offset variation along the orbit for selected wavenumbers in the different spectral bands in November 2003 (FR mode). The position of the offset measurements within the orbit is represented in terms of latitude. $0°$ represents the ascending equator crossing, $90°$ represents the north pole, $180°$ the descending equator crossing and $270°$ the south pole. The offset measurements are always taken at the same latitude positions. In order to reduce

the noise level, the offset spectra of 15 orbits have been co-added for each latitude position (i.e. 90 spectra per measurement point, since 6 sweeps (3 forward and 3 reverse) are taken per offset measurement). The variation between two subsequent offset measurements (i.e. between two data points in Fig. 9) is below $2\,\mathrm{nWcm^{-2}sr^{-1}cm}$ in band A and even lower in the other spectral bands. In OR mode, where the time span between two offset calibration measurements is larger, the variation is below about $4\,\mathrm{nWcm^{-2}sr^{-1}cm}$ and still below the offset error due to noise."

*p13 l26 – 28: Did you assume 91 % modulation efficiency for all bands? Or did you derive it using the DC-zero method for all detectors but B1 and B2? Please clarify.*

This assumption is valid for all non-linear channels. We will clarify this by writing:

"Therefore the modulation efficiency is estimated from the optical specifications and instrument properties to be 91 % in all non-linear channels (Kleinert et al., 2015). This is based on the assumption that the instrument is well aligned and the modulation efficiency is rather wavenumber independent in the relevant spectral range of 685 to $1500\ \mathrm{cm^{-1}}$."

*p13 l31: And what about bands AB, C, D?*

With "B and A bands" we meant the channels B1 and B2 as well as the channels A1 and A2. In order to make this clear and following a suggestion of reviewer 1, we will rephrase the text and write:

"There are three main sources of uncertainty for the determination of the non-linearity: (1) the assumption, that the detector curve is characterized by a 3rd order polynomial for channels A1 and A2 and by a 2nd order polynomial for channels B1 and B2, ..."

Bands C and D are not subject to non-linearity correction.

*p18 Figure 12: Is my assumption correct that altitude level 1 is low (about 10 km) and altitude level 26 is high (about 65 km) altitude? Please provide a description in the caption.*

We will add the corresponding information in the caption:

"Altitude 26 corresponds to about 67 km, altitude 1 to about 7 km."

*Please consider a change of the color scale to blue/green for excellent agreement (ratio close to 1) and orange/red for slightly reduced agreement (ratio smaller than 1).*

Many people associate red with high and blue with low values, therefore we want to keep the color scale as it is. Instead, we will invert the ratio, i.e. we will show A2 / A1 instead of A1 / A2. Then the reduced agreement will be represented by orange / red colors. Furthermore this representation is in better agreement with Fig. 11, where also the ratio of A2 / A1 is shown.

*p20 l1 – 3, l6-15: Do you have any idea what is causing the day-night difference, latitudinal and seasonal variation of the offset?*

No. We have searched for correlation with instrument temperatures and cloud cover but we did not find a significant correlation. We will add a corresponding sentence at the end of section 6.2 in the text:

"Also the day/night variation as well as the seasonal, latitude dependent variation of the offset cannot be explained with known instrument effects, but the observed offset variation gives an impression of the expected offset error and its variation."

*p22 l6: Please explain why the spectral accuracy is given like a mixing ratio in ppm. I'd rather expected values in cm-1 as you provide in line 7.*

The spectral accuracy is driven by the exact knowledge of the laser wavelength, and an error in the laser wavelength corresponds to a scaling error of the spectral axis. Therefore the absolute error (in $cm^{-1}$) is wavenumber dependent while the relative error is valid for the whole spectral range. The unit ppm is a general relative unit which is widely used for mixing ratios but is also very common in other fields. We think that 0.14 ppm is easier to read than 0.000014 %, which would be equivalent.

*p23 l12: Which standard atmosphere did you use (the U.S. Standard Atmosphere or any other)? Is this atmosphere anywhere available?*

The atmosphere used is basically the U.S. Standard Atmosphere with some minor modifications.

*Technical comments*

All technical comments will be addressed in the text.

*Text font and unit font seem to be different throughout the manuscript e.g p1 l13.*

This is the outcome of the Copernicus template.